# The Bifunctional Effects of Lactoferrin (LFcinB11) in Inhibiting Neural Cell Adhesive Molecule (NCAM) Polysialylation and the Release of Neutrophil Extracellular Traps (NETs)

**DOI:** 10.3390/ijms25094641

**Published:** 2024-04-24

**Authors:** Bo Lu, Si-Ming Liao, Shi-Jie Liang, Li-Xin Peng, Jian-Xiu Li, Xue-Hui Liu, Ri-Bo Huang, Guo-Ping Zhou

**Affiliations:** 1National Key Laboratory of Non-Food Biomass Energy Technology, National Engineering Research Center for Non-Food Biorefinery, Institute of Biological Science and Technology, Guangxi Academy of Sciences, 98 Daling Road, Nanning 530007, China; lubo@gxas.cn (B.L.); liaosiming@gxas.cn (S.-M.L.); liangshijie@gxas.cn (S.-J.L.); penglixin@gxas.cn (L.-X.P.); jianxiuli@gxas.cn (J.-X.L.); 2Institute of Biophysics, Chinese Academy of Sciences, Beijing 100101, China; xhliu@ibp.ac.cn; 3Rocky Mount Life Sciences Institute, Rocky Mount, NC 27804, USA

**Keywords:** migration of tumor cells, neuronal cell adhesion molecule, polysialic acid, polysialyltransferase, polysialyltransferase domain, lactoferrin (LFcinB11), neutrophil extracellular traps (NETs), NMR spectroscopy, chemical shift perturbation

## Abstract

The expression of polysialic acid (polySia) on the neuronal cell adhesion molecule (NCAM) is called NCAM-polysialylation, which is strongly related to the migration and invasion of tumor cells and aggressive clinical status. Thus, it is important to select a proper drug to block tumor cell migration during clinical treatment. In this study, we proposed that lactoferrin (LFcinB11) may be a better candidate for inhibiting NCAM polysialylation when compared with CMP and low-molecular-weight heparin (LMWH), which were determined based on our NMR studies. Furthermore, neutrophil extracellular traps (NETs) represent the most dramatic stage in the cell death process, and the release of NETs is related to the pathogenesis of autoimmune and inflammatory disorders, with proposed involvement in glomerulonephritis, chronic lung disease, sepsis, and vascular disorders. In this study, the molecular mechanisms involved in the inhibition of NET release using LFcinB11 as an inhibitor were also determined. Based on these results, LFcinB11 is proposed as being a bifunctional inhibitor for inhibiting both NCAM polysialylation and the release of NETs.

## 1. Introduction

The expression of polysialic acid (polySia) on neuronal cell adhesion molecule (NCAM) [1,2,3,4,5,6,7] is called NCAM polysialylation, which is related to cancer cell migration through interactions between polysialyltransferases (polySTs) and CMP-Sia and polyST and polysialic acid (polySia) [8,9]. More specifically, these interactions actually involve the direct bindings of the Polysialyltransferase Domain (PSTD) to CMP-Sia and PSTD to polySia [10,11,12,13,14,15]. PSTD is a polybasic motif of 32 amino acids in two polySTs: ST8SiaIV and ST8SiaII [16].

The inhibition of posttranslational modifications (PTMs) is related to a number of diseases, such as cancer and nervous and cardiovascular system diseases. One of the latest advances in research on PTMs is the inhibition of the polysialylation of neuronal cell adhesion molecule (NCAM) [17,18], which is strongly related to the migration and invasion of tumor cells and is associated with aggressive, metastatic disease and poor clinical prognosis in the clinic due to the formation of polysialic acid (poly-Sia) on the surface of NCAM [19,20,21,22].

It has been known that NCAM-polySia expression on cancer cells is catalyzed by two polysialyltransferases (polySTs), ST8SiaIV and ST8SiaII; specifically, two polybasic motifs, Polybasic Region (PBR) and Polysialyltransferase Domain (PSTD) within each polyST, have been found to be critically important for polyST activity based on recent mutation and molecular modeling analyses [16,23]. Thus, the intermolecular interactions of PBR-NCAM, PSTD-polySia, and PSTD-(CMP-sialic acid) have been suggested during NCAM polysialylation and have been tested in more recent NMR studies [14]. Furthermore, a modulation model of NCAM polysialylation and cell migration has been proposed by incorporating the intramolecular interaction of PBR-PSTD into the above intermolecular interaction [14]. This model has been further supported using Chou’s wenxiang diagram method [24,25,26,27,28].

Two inhibitors of NCAM polysialylation, low-molecular-weight *heparin* (LMWH) and cytidine monophosphate (CMP), have been proposed in drug research and have shown help in developments related to tumor-targeted polysialyltranseferases based on the above modulation model of NCAM polysialylation [16,23].

polyST activity decreased by 93% due to the binding of PSTD-heparin, according to previous in vitro experiments [15]. This finding suggests that heparin is an effective inhibitor of NCAM polysialylation. In recent studies, the binding affinity between the PSTD and three different types of heparin, unfractionated heparin (UFH), heparin tetrasaccharide (DP4), and low-molecular-weight LMWH have been compared using NMR spectroscopy, and it has been further verified that LMWH is the strongest inhibitor among these three types of heparin [21]. However, the use of heparin should be carefully considered, as previous reports have indicated that intracerebral hemorrhages in patients were related to heparin intake [29].

Another inhibitor, cytidine monophosphate (CMP), has also been shown to partially inhibit polysialylation when CMP-Sia and polySia co-exist in solution in recent NMR studies. CMP-Sia may play a role in reducing the extent of accumulation of polySia chains on the PSTD and may be beneficial in inhibiting polysialylation [30]. However, CMP could not inhibit the PSTD–polySia interactions [30].

Different ways to block cancer initiation and progression are of great interest. Many pathways influencing the life of cancer cells may be targeted, including escape from the site of inception or invasion into the metastatic site or death of the cancer cell, such as through necroptosis. One pathway of cell death relevant to some cancers concerns neutrophil extracellular traps (NETs) [31,32]. The activity of NETs may increase during neoplasia, and the rate of cancer growth increases by generating greater amounts of degraded cells. These activations are inhibited by lactoferrin (LF), an iron-binding glycoprotein composed of 49 amino acids. LF has antimicrobial, antiviral, antitumor, and immunological activities [31,32].

It has been described that the LF-derived peptide lactoferricin (LFcin) belongs to the family of antimicrobial peptides [33,34]. LFcin peptide interacts with polySia, and the binding of lactoferrin to polySia is mediated by LFcin, which is included in the N-terminal domain of lactoferrin. The binding between LFcin and polySia was confirmed through the use of native agarose gel electrophoresis [34]. Remarkably, as an antimicrobial peptide, the protective activity of LFcin was not impacted [35,36]. However, previous studies were all conducted at the biochemical, isolated protein level. No works with cultured cell growth models or whole-animal neoplasia models have been undertaken. Before more studies on in vitro work are considered, it is necessary to establish the importance of polySia-LF interaction in vivo in terms of functional effects. The vast majority of reports published to date are either test tube type or use cell cultures. Both are very good in terms of elucidating mechanisms of action but are derivative from a healthcare perspective as they can only be effectively modeled in vivo in intact animals.

In recent studies, a 11-residual peptide (RRWQWRMKKLG) from the N-teminus [32] of LF was designed LFcinB11, which also has similar antimicrobial activities to those seen in bovine lactoferricin (BLFC) [33,34].

In this study, our aim was to determine whether LFcinB11 can also inhibit NCAM polysialylation. If so, what are the molecular details of this interaction? Previous studies have proposed that the release of neutrophil extracellular traps (NETs) could be inhibited through the interaction between polySia and LFcin in in vitro experiments [31,32,33]. In this study, the molecular details of both LFcin-PSTD and LFcin-polySia interactions were explored for the first time using NMR spectra. The results of this study indicate that LFcinB11 may play a bifunctional role in inhibiting the formation of NETs and NCAM polysialylation. These findings may provide a deep insight into drug design for cancer cell inhibition and migration.

## 2. Results

### 2.1. CD Data

As shown in Figure 1, our CD spectra display that the α-helical content of the PSTD contains 23.0% in the absence of any ligands. The helical contents decreased to 16.4% after adding a mixture of 40 μM (CMP-Sia) and 4 μM polySia and further decreased to 14.8% after the addition of LFcinB11. These results suggest the helices in the PSTD were unwound due to the addition of the mixture of CMP-Sia and polySia and were further unwound after the addition of LFcinB11. The decreased helical contents in the PSTD indicate its conformational change, verifying that the PSTD not only interacts with the mixture of CMP-Sia and polySia, as shown in previous studies [10,11,12,13,14], but also suggests an interaction between the PSTD and LFcinB11. However, the difference between 16.4% and 14.8% is only 1.6%. This means that the helical structure of the PSTD was basically stable after LFcinB11 was added to the sample. A possible explanation is that there is an interaction between LFcinB11 and polySia. This may be due to the fact that the interaction may decrease the extent of helical unwinding in the PSTD.

### 2.2. NMR Results (a)

In order to verify the interaction between the PSTD and LFcinB11, 2D-HSQC experiments involving the mixtures of LFcinB11 with different concentration and the PSTD were carried out. In addition, the PSTD-LFcinB11 interaction was also tested.

#### 2.2.1. The Interaction between the PSTD and 20 μM LFcinB11

In this study, the overlaid HSQC spectra of the PSTD for the PSTD-(20 μM LFcinB11) interaction showed that significant changes in chemical shift could be found in eight residues: K246, K250, V251, R252, T253, S257, V273, and I275 (Table 1 and Figure 2A); most residues are located on the binding region of CMP-Sia (K246-L258) (Table 2), except for two residues: V273 and I275 (Figure 2A). In addition, the CSP values in this range (K246-L258) for the PSTD-20 μM LFcinB11 interaction are less than that for the PSTD-(CMP-Sia) interaction (Figure 2). These results indicate that 20 μM LFcin11 could not inhibit the PSTD-(CMP-Sia) interaction. Similarly, the CSPs values in the polySia binding region of the PSTD (A263-N271) are also smaller compared to the PSTD-polySia interaction (Figure 3).

#### 2.2.2. The Interaction between the PSTD and 40 μM LFcinB11

When the LFcinB11 concentration increased to 40 μM, significant changes in chemical shift were found in 19 residues: K246, K248, K250, V251, R252, T253, A254, Y255, S257, L258, V264, R265, Y267, W268, L269, V273, I275, R277, and S279; there were only six residues (R259, V260, I261, H262, A263, and K272) where no change in chemical shift was observed (Table 1 and Figure 2B). The CSPs for PSTD-40 μM LFcinB11 interaction were larger than that for the PSTD-(CMP-Sia) and PSTD-20 μM LFcinB11 interactions (Figure 3A) in the CMP-Sia binding region, indicating the PSTD-(CMP-Sia) interaction could be inhibited by 40 μM LFcin. In addition, most CSPs for PSTD-40 μM LFcin interaction and the PSTD-polySia interaction are very close in the polySia binding region (Figure 3B). However, the CSPs for the former are larger than that for the latter at residues L269 and N271 (Figure 3B). Thus, this suggests that the PSTD-polySia interaction could be inhibited with a LFcinB11 concentration of more than 40 μM.

#### 2.2.3. The Interaction between the PSTD and 60 μM LFcinB11

When the LFcin11 concentration was increased to 60 μM, significant changes in chemical shift were found in most residues according to the overlaid HSQC spectra (Figure 2C), and there are only three residues (R259, V260, and A263) where no change in chemical shift was observed (Table 1 and Figure 2C). The CSPs for PSTD-60 μM LFcinB11 interaction are larger than that for PSTD-(CMP-Sia) interaction (Figure 3A) in the CMP-Sia binding region, indicating that PSTD-(CMP-Sia) interaction could be inhibited by 60 μM LFcinB11. In addition, the CSPs for PSTD-60 μM LFcinB11 interaction are also larger than that for PSTD-polySia interactions in the polySia binding region (Figure 3B); thus, this further suggests that PSTD-polySia interactions could be inhibited through the use of 60 μM LFcinB11.

#### 2.2.4. The Interaction between the PSTD and 80 μM LFcinB11

When the LFcinB11 concentration was increased to 80 μM, almost all residues of the PSTD changed in relation to their chemical shift (Figure 2D), and the CSP of each residue was larger than that observed for the interactions between the PSTD and 60 μM LFcin. In the CMP-Sia binding region of PSTD-(CMP-Sia) interactions, the maximum CSP is 0.151, which is larger than that observed for the PSTD-60 μM LF, and in the polySia binding region, the maximum CSP is 0.109, which is also much larger than that observed for PSTD-60 μM LFcinB11 (Table 2). These results indicate that the CSPs in both CMP-Sia and polySia binding regions of the PSTD increased with increasing LF concentrations.

#### 2.2.5. The Interactions between LFcinB11 and polySia

In order to determine the interaction between polySia and LFcinB11, the overlaid 2D 1H-15N HSQC spectra were examined using our NMR spectrometer. As shown in Figure 4, no chemical shift was detected except for residue 11 G after polySia and LFcinB11 were mixed. However, the peak intensities of almost all residues were significantly decreased, thus suggesting the interaction between polySia and LFcinB11 through the formation of LFcinB11-polySia aggregates.

## 3. Discussion

To date, neither 3D X-ray not NMR structures of polySTs have been reported due to the existence of many hydrophobic residues in polySTs, which are situated in the membrane environment [13,14]. However, a 3D structure of the PSTD peptide in solution, an active site in ST8Sia IV, has been obtained based on our NMR studies [10,11,12,13,14]. Thus, the following hypothesis has been proposed: interactions between the PSTD and ligands, such as CMP-Sia, polySia, or any possible inhibitors, may correspond to interactions between polyST and these ligands. This is an efficient research strategy and methodology for studying biological problems using biophysical and NMR structural biology. The above hypothesis has been successfully tested in recent NMR studies [12,21,30].

The above CD spectra qualitatively demonstrate the possible interactions between the PSTD and LfcinB11. More details on the interactions between the PSTD and Lfcin11 were provided by our experimental NMR results.

The polysialylation of the trimer of α-2,8-linked sialic acid (triSia) was inhibited by cytidine monophosphate (CMP) in the presence of ST8SiaII and CMP-Neu5Ac (CMP-Sia) based on in vitro experiments [30]. More recent studies have verified that PSTD-(CMP-Sia) could be inhibited by CMP, but PSTD-polySia binding could not be inhibited by CMP, even in a mixture of CMP-Sia, polySia, and the PSTD based on our NMR data [30].

There are two binding regions for CMP-Sia in the PSTD; one is in the residue range of K246-L258, and the other one is in the range of Y267-R277 [30]. The former is also the binding region for CMP, and the latter is covered by the CMP-PSTD binding region (V264-K276) [30]. In this study, the CSP values for the PSTD-LFcinB11 interactions are larger than that for PSTD-CMP and the PSTD-polySia interactions when using a LFcinB11 concentration of at least 40 μM (Figure 5). These results indicate that LFcinB11 is more powerful in inhibiting both the PSTD-(CMP-Sia) and PSTD-polySia interactions than CMP.

Previous NMR studies have indicated that LMWH is an effective inhibitor of NCAM polysialylation. Twelve residues, N247, V251, R252, T253, S257, R265, Y267, W268, L269, V273, I275, and K276 in the PSTD were discovered to be the binding sites of LMWH; they were mainly located on the long α-helix of the PSTD and the short three-residue loop of the C-terminal PSTD [23]. The range of LNWH binding to the PSTD is almost the same as that of the LF (Figure 6). As shown in Figure 6A and Table 2, the CSPs of the PSTD for the PSTD-LMWH (80 μM) interaction are larger than that for PSTD-(CMP-Sia) interaction, indicating that PSTD-(CMP-Sia) binding could be inhibited by 80 μM LMWH. However, when only taking 40 μM LF, both PSTD-(CMP-Sia) interaction and PSTD-polySia interaction could be inhibited (Figure 6B). In addition, as an inhibitor, LFcinB11 may be safer than LMWH due to the fact that intracerebral hemorrhage is related to related to heparin intake [29].

LFcinB11 can not only interact with the PSTD to inhibit PSTD-(CMP-Sia) and PSTD-polySia interactions but can also directly interact with polySia (Figure 4). This NMR result is consistent with the results obtained when using in vitro experiments [32,33,34] and implies that the major contribution of the interaction between LF and polySia is from the N-terminal residues of LF, particularly in the LFcinB11 domain.

## 4. Materials and Methods

### 4.1. Material Sources

The PSTD (246K-277R) is a 32 amino acid sequence peptide from the ST8Sia IV molecule, which was derived from human cells [15]. In order to obtain more accurate 3D structural information through the use of NMR spectroscopy, one amino acid (245L) and two amino acids (278P and 279S) from the ST8Sia IV sequence were added to the N- and C- terminals of the PSTD, respectively [12,13]. Thus, a 35 amino acid sequence peptide sample containing the PSTD was synthesized as follows: “245LKNKLKVRTAYPSLRLIHAVRGYWLTNKVPIKRPS279”. Herein, the PSTD sequence is underlined. This intact peptide sample was chemically synthesized via automated solid-phase synthesis using the F-MOC-protection strategy and purified through the use of HPLC (GenScript, Nanjing, China). Its molecular weight was determined to be 4117.95 and its purity was established to be 99.36%.

The LFcinB11 peptide was purchased from BACHEM with the following amino sequence RRWQWRMKKLG; it had a relative molecular mass of 1544.8. PolySia was purchased from Santa Cruz Biotechnology.

### 4.2. Circular Dichroism (CD) Spectroscopy

The concentrations of the 35 amino acid-PSTD peptide and LFcinB11 in 20 mM phosphate buffer (pH 6.7) with 25% tetrafluoroethylene (TFE) were 8.0 μM and 400 μM, respectively. The methods used to measure and record the CD spectra were the same as those described in previous articles [23,30].

### 4.3. NMR Sample Preparation

The 35 amino acid peptide containing the PSTD was prepared as described above in a 20 mM phosphate buffer containing 25% TFE. Chemical shifts were referenced with respect to 2-dimethyl-2-silapentane-5-sulfonic acid (DSS), which was used as the internal standard.

For both 1-D and 2-D NMR experiments, the concentration of the PSTD peptide in the absence or presence of LFcinB11 was 2.0 mM. The concentrations of LFcinB11 in the presence of the PSTD were all 20, 40, 60, and 80 μM, respectively. For 2D NMR experiments of polySia-LBcinB11 interaction, the concentrations of polySia and LFcinB11 were all 50 μM.

All of the NMR samples were dissolved in 25% TFE (*v*/*v*), 10% D_2_O (*v*/*v*), and 65% (*v*/*v*) 20 mM phosphate buffer (pH 6.7). Following this, 2-Dimethyl-2-silapentane-5-sulfonic acid (DSS) was added to all samples to serve as a reference standard.

### 4.4. NMR Spectroscopic Methods

NMR spectroscopy is a *powerful* tool for studying biomolecule–protein (DNA) or protein–ligand interactions [36,37,38,39,40,41,42,43,44,45,46,47,48,49]. All of the NMR spectra were recorded at 298 K using an Agilent DD2 800 MHz spectrometer equipped with a cold probe in the NMR laboratory at the Guangxi Academy of Sciences. Water resonance was suppressed using pre-saturation. The NOESY mixing times were set at 300 msec, while the TOCSY experiments were recorded with mixing times of 80 msec [31,50]. All of the chemical shifts were referenced to the internal DSS signal set to 0.00 ppm for protons and indirectly for carbon and nitrogen [21,30]. The data were typically apodized with a shifted sine bell window function and zero-filled to double the data points in F1, prior to being Fourier-transformed. NMRPipe [21,30]. CcpNmr [21] was used to process the data and perform the spectral analysis. Spin system identification and sequential assignment of individual resonances were carried out using a combination of TOCSY and NOESY spectra, as previously described [21,30], and subsequently coupled with an analysis of 1H-15N and ^1^H-^13^C HSQC for overlapping resonances. In order to identify and characterize the specificity of PSTD–ligand binding, the chemical shift perturbation (CSP) of each amino acid in the PSTD was calculated using the following formula:CSP = [(D^2^_NH_ + (D_N_/5)^2^)/2]^1/2^(1)
where D_N_ and D_NH_ represent the changes in ^15^N and ^1^H chemical shifts, respectively, upon ligand binding [50,51,52,53,54,55,56,57,58].

## 5. Conclusions

Our results indicate that LFcin11 is a more powerful inhibitor than LMWF and CMP and can be safely used. Furthermore, the bifunctional effects of LFcinB11 are proposed, i.e., LFcinB11 can not only inhibit NCAM polysialylation through PSTD-LFcinB11 interaction but can also inhibit the formation of neutrophil extracellular traps (NETs), a network of extracellular strings of DNA that bind to pathogenic microbes [59,60,61,62,63,64,65,66,67,68,69,70]. Due to the fact that the role of NETs in promoting tumor metastasis formation could be blocked via the addition of LFcin B11 [71,72,73,74,75,76,77,78,79,80,81], a bifunctional effect of LFcinB11 has been proposed in this study. In future studies, we will further study the molecular mechanism of the interactions between polySia and LFcinB11 to understand how LFcin blocks tumor metastasis formation related to the formation of NETs.

## Figures and Tables

**Figure 1 ijms-25-04641-f001:**
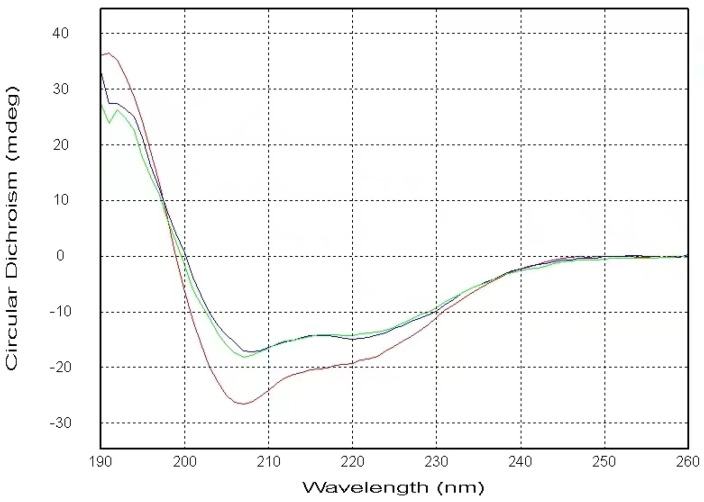
The figure shows the CD spectra of the PSTD in the absence (red) and presence of the mixture of CMP-Sia and polySia (green) and the mixture of CMP-Sia, polySia, and LFcinB11 (blue). The helical contents of these three CD spectra of the PSTD are 23%, 16.4%, and 14.8%, respectively.

**Figure 2 ijms-25-04641-f002:**
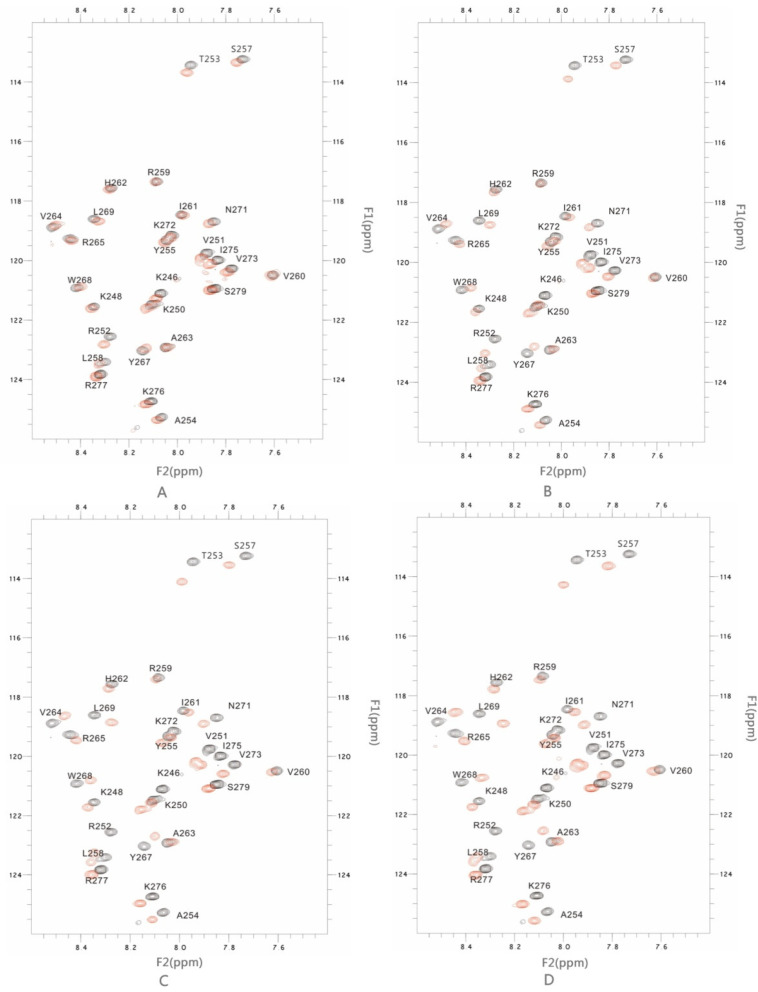
The figure shows the overlaid 1H-15N HSQC spectra of the 2 mM PSTD in the absence and presence of 20 μM LFcinB11 (**A**), 40 μM LFcinB11 (**B**), 60 μM LFcinB11 (**C**) and 80 μM LFcinB11 (**D**), respectively.

**Figure 3 ijms-25-04641-f003:**
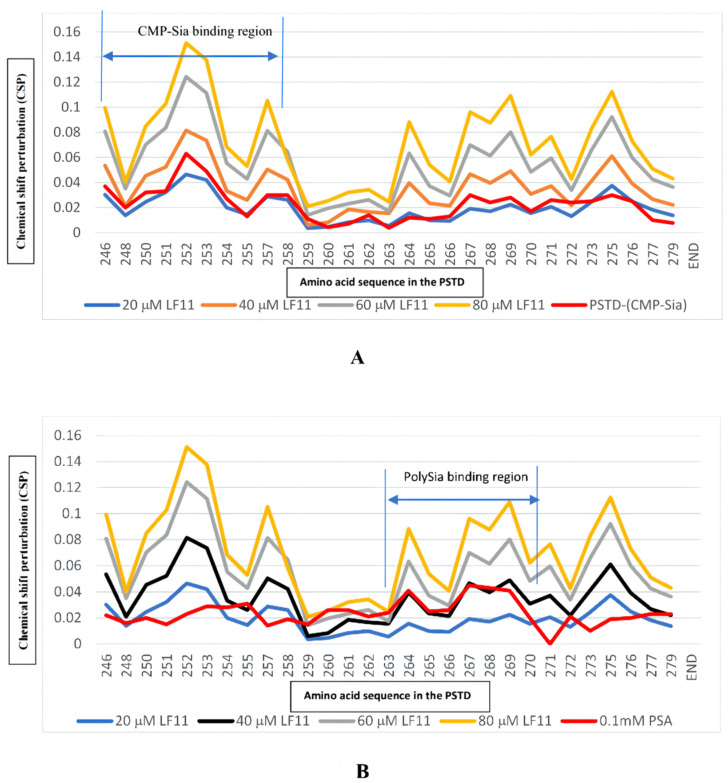
The figure shows the chemical shift perturbations (CSPs) of the PSTD when the PSTD interacted with 20 μM LFcinB11 (blue), 40 μM LFcinB11 (orange), 60 μM LFcinB11 (gray), and 1 mM CMP--Sia (red), respectively (**A**); chemical shift perturbations (CSPs) of the PSTD when the PSTD interacted with 20 μM LFcinB11(blue), 40 μM LFcinB11 (black), 60 μM LFcinB11 (gray), 80 μM LFcinB11 (orange), and 0.1 mM polySia (red), respectively (**B**).

**Figure 4 ijms-25-04641-f004:**
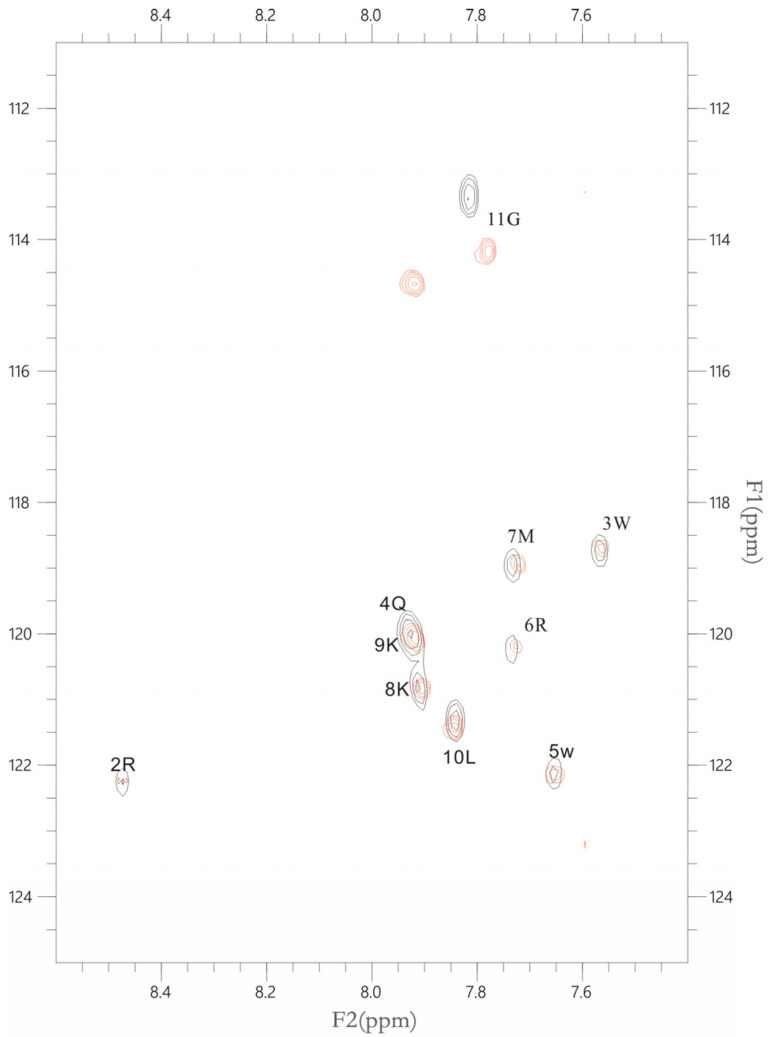
The figure shows the overlaid 1H-15N HSQC spectra of 50 μM LFcin11 in the absence (black) and presence of 50 μM polySia (red), respectively.

**Figure 5 ijms-25-04641-f005:**
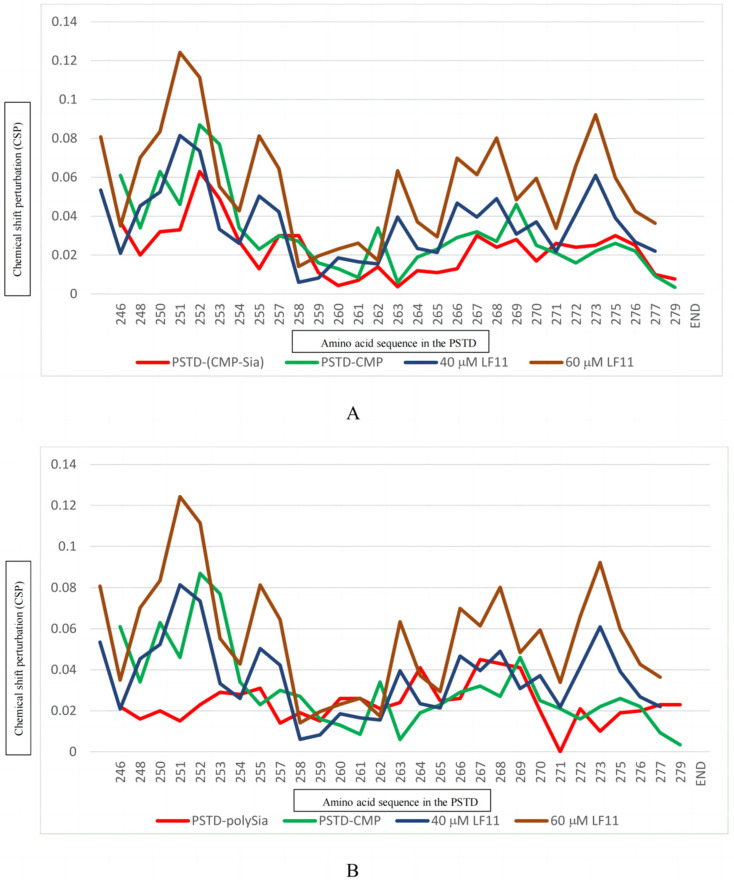
The figure shows the chemical shift perturbations (CSPs) of the PSTD when interacting with 1 mM CMP-Sia, 80 μM CMP, 40 μM, and 60 μM LFcinB11, respectively (**A**), and the CSPs of the PSTD when interacting with 0.1 mM PSA, 1 mM CMP, 40 μM, and 60 μM LFcinB11, respectively (**B**).

**Figure 6 ijms-25-04641-f006:**
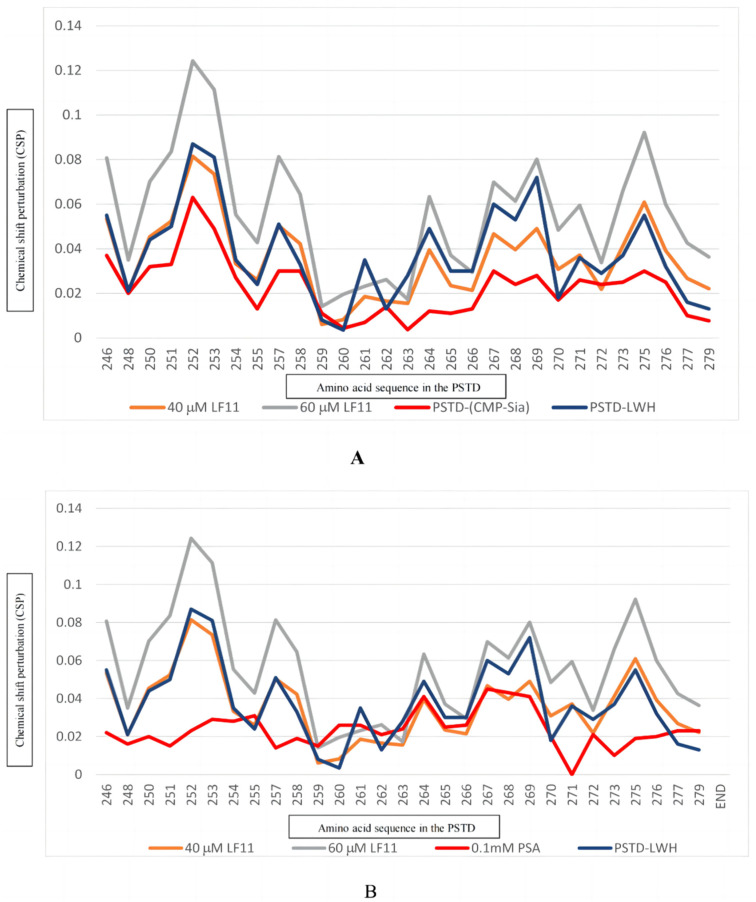
The figure shows the chemical shift perturbations (CSPs) of the PSTD when interacting with 1 mM CMP-Sia, 80 μM heparin LMWH, 40 μM, and 60 μM LFcinB11, respectively (**A**), and the CSPs of the PSTD when interacting with 0.1 mM PSA, 80 μM heparin LWH, and 40 μM and 60 μM LFcinB11, respectively (**B**).

**Table 1 ijms-25-04641-t001:** The table shows the effect of different lactoferrin concentrations (20 μM, 40 μM, 60 μM, and 80 μM) on the chemical shift of the residues for the PSTD-LFcinB11 interaction based on the data from Figure 2 and Figure 3.

LFcinB11 Concentration Interacting with the PSTD	Residues in the PSTD That Do Not Change in terms of Chemical Shift	Residues in the PSTD That Changed in relation to Chemical Shift
20 μM	17 residues (K248, A254, Y255, L258, R259, V260, I261, H262, A263, V264, R265, Y267, W268, L269, K272, K276, and S279)	8 residues (K246, K250, V251, R252, T253, S257, V273, and I275)
40 μM	6 residues (R259, V260, I261, H262, A263, and K272)	19 residues (K246, K248, K250, V251, R252, T253, A254, Y255, S257, L258, V264, R265, Y267, W268, L269, V273, I275, R277, and S279)
60 μM	3 residues (R259, V260, and A263)	22 residues
80 μM	0 residues	25 residues

**Table 2 ijms-25-04641-t002:** The table shows the binding regions of CMP-Sia and polySia for the different ligands on the PSTD. The maximum CSPs in each binding region are compared with the maximum CSPs for PSTD-(CMP-Sia) and PSTD-polySia interactions, respectively. All CSPs were obtained based on current and previous 2D 1H-15N HSQC experiments [21,30].

Ligands Binding to the PSTD	The Maximum CSPs in CMP-Sia Binding Region (K246-L258)	The Maximum CSPs in polySia Binding Region (A263-R271)
CMP-Sia	0.063	0.030
polySia	0.031	0.045
Heparin LMWH (80 μM)	0.087	0.072
CMP (1 mM)	0.087	0.046
LFcinB11 (20 μM)	0.047	0.038
LFcinB11 (40 μM)	0.081	0.061
LFcinB11 (60 μM)	0.124	0.092
LFcinB11 (80 μM)	0.151	0.109

## Data Availability

The original contributions presented in the study are included in the article, further inquiries can be directed to the corresponding author/s.

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
