# Peer review of "The Bifunctional Effects of Lactoferrin (LFcinB11) in Inhibiting Neural Cell Adhesive Molecule (NCAM) Polysialylation and the Release of Neutrophil Extracellular Traps (NETs)"

_ijms, 2024, doi:10.3390/ijms25094641_

Round 1

Reviewer 1 Report

Comments and Suggestions for Authors

Content suggestions:

1.         I would like to kindly ask the Authors whether they used protein derived from humans, as this is not mentioned in the manuscript exactly...?

2.         Do the Authors have any information about the effectiveness of the particular types of LMWH in inhibiting the migration and invasion of tumor cells (dalteparin, nadroparin, enoxaparin, tinzaparin...) ?

Reviewer 2 Report

Comments and Suggestions for Authors

            Different ways to block cancer initiation and progression are of great interest.  Many pathways influencing the life of cancer cells may be targets, including escape from site of inception or invasion into metastatic site, or death of the cancer cell such as by necroptosis.  One pathway of cell death relevant in some cancers are neutrophil extracellular traps (NET).  The activity of NET may increase during neoplasias and by genereating greater amounts of degraded cells the rate of cancer growth increases. These activations are inhibited by lactoferrin binding.  I hope this is correct, the model is a challenge to decipher clearly.  There is only biochemical interaction data as for the X-ray and NMR spectra which are the first time these are reported.  The authors must be credited they reference and comment on data from a 2019 publication on lactoferrin and polySia interaction.  The present study investigations are all at the biochemical, isolated protein level.  No work with cultured cell growth models or whole animal neoplasia models are investigated.  Before more studies on in vitro work is considered, it is important to establish the importance of polySia/lactoferrin interactions in vivo through functional effects.  The vast majority of reports on  NET are either test tube type or using cell cultures.  Both are very good to elucidate mechanisms of action, but are derivative from a healcth concern that can only be modeled in vivo in intact animals. Imagination and care to build such a  NET/cancer experiment but it is pivotal.  One other manner to enhance the X-ray and NMR data would be to determine how, meaning signaling pathways that lactoferrin uses to induced NET.  Inclusion of data on lactoferrin reduction of cancer growth, which has been reported in the literature, would add to the studies, particularly if the role of NET for lactoferrin could be dissected out whch gene knockdown systems may allow.

Comments on the Quality of English Language

In more than a few occasions, unusual English meaning is used.  Meanings are still clear, but a native English speaker would say these statements differently.
